# Study on Crack Propagation of Rock Bridge in Rock-like Material with Fractures under Compression Loading with Sudden Change Rate

Xuanqi Huang [1,*], Wen Wan [1,2,3], Min Wang [2], Yu Zhou [2], Jie Liu [4] and Wei Chen [4]

[1] School of Civil Engineering, Hunan University of Science and Technology, Xiangtan 411201, China
[2] School of Resource, Environment and Safety Engineering, Hunan University of Science and Technology, Xiangtan 411201, China
[3] Work Safety Key Laboratory on Prevention and Control of Gas and Roof Disasters for Southern Coal Mines, Hunan University of Science and Technology, Xiangtan 411201, China
[4] Architectural Engineering Institute, Hunan Institute of Engineering, Xiangtan 411101, China
[*] Correspondence: 21010201015@mail.hnust.edu.cn

**Abstract:** In order to study the influence of sudden change of loading rate on crack propagation and failure mode of rock bridge of fractured rock mass, the specimens used in this paper were rock-like materials containing two prefabricated fractures. The mechanical properties and the failure mode of the specimens under different constant loading rates and sudden changes in loading rate were tested. The photographic monitoring and acoustic emission were carried out at the same time. The results show that: (1) In the process of sudden change loading, the specimen shows the *characteristics* of approximate elastic deformation, and the stress-strain curve after sudden loading is similar to the corresponding stress-strain curve at the corresponding constant loading rate. (2) Combined with acoustic emission detection, it is found that when the loading rate is not much different before the mutation, the loading rates is the same after the mutation and the stress-strain curve of the specimen is similar. In the low-speed loading stage, the acoustic emission count is generally low, while in the high-speed loading stage, the acoustic emission count is generally high. The sudden change from low-speed loading to high-speed loading easily induces stress drop, resulting in crack generation and increase of acoustic emission count. (3) The rate of the specimen during the crack development period plays a decisive role in the failure mode of the specimen. Before the sudden change of loading rate, the low-speed loading within a certain range has little effect on the specimen. When the high-speed loading is carried out when the stress is low in the early stage of the mutation, the acoustic emission count of the specimen is high, which will cause some damage inside the specimen. As a result, even if the rate is the same after the mutation, the final peak stress and failure mode may be different.

**Keywords:** mechanical damage; micro mechanism; sudden change loading rate; loading rate; crack propagation

## 1. Introduction

There are a large number of natural joints, cracks, structural planes, and other weak surfaces in natural rock mass [1]. These weak surfaces play a significant role in the stability of rock masses during the construction of tunnels, mining roadways, dams, slopes, and underground civil air defense projects. The stability of the rock mass directly affects the construction progress, as well as the safety during and after its use. Due to the continuous progression of surface construction, underground excavation, and mining, the load on the rock mass is subjected to varying loads instead of a constant value. Studies have shown that many rocks demonstrate an obvious loading rate effect [2]. Changes in loading rate applied to the rock mass have significant influence on the weak surface materials, creating primary or secondary cracks within the rock mass, which may continuously accelerate or slow

down to close, crack initiation, propagation, coalescence, and even lead to instability and failure of rock materials [3]. In severe cases, the change of load may lead to the instability and failure of rock mass, which not only brings huge economic losses to the project, but also poses a threat to the life safety of relevant staff. Therefore, it is of great engineering significance to study the mechanical properties, crack propagation, and failure modes of fractured rock materials under sudden change in loading rate uniaxial compression, which helps to assess the damage to the rock.

In recent years, both domestic and foreign scholars have conducted a lot of research on the loading rate effect of rocks by conducting laboratory tests. Li Fulin et al. [4] found that mudstone had obvious loading rate change effect and showed isotach viscosity behavior. Yu Liqiang et al. [2] studied that the loading rate had an effect on the crack propagation and failure mode of the rock specimen. With the increase of the loading rate, the crack generation types changed from wing crack to tensile crack, and no longer transformed to other crack types. Wang Xiaoran et al. [5] discovered that there was obvious loading rate effect on crack propagation and acoustic emission response behavior of fractured sandstone. He Song et al. [6] took clay rocks as test objects to study the loading rate effect of rocks. The results revealed that clay rocks presented a tendency to approximate linear change with the increase of uniaxial compression loading rate. Qi et al. [7] researched that the peak strength of some rocks did not increase with the increase of loading rate.

In the process of rock loading, the acoustic emission (AE) would have occurred in the form of stress waves when the internal crystals were dislocated or fractured. Therefore, acoustic emission technology is an effective way to monitor the failure of rock mass [8]. Eberhardt et al. [9] studied the failure process of granite using AE technology and proposed that the occurrence of significant AE events or the surge of AE characteristic parameters often corresponded to the initiation of microcracks. Moradian et al. [10] compared acoustic emission signals with stress–strain curves of rocks, founding that acoustic emission ringing and its energy characteristics could better reflect the activity of fracture events inside rocks and the elastic strain energy released. Xuepeng Song et al. [11] analyzed the mechanical properties and mesoscopic acoustic emission characteristics of prefabricated fracture cemented paste backfill under different loading rates, and concluded that the temporal and spatial distribution characteristics of AE corresponded to the crack evolution trend. Zhao Xingdong et al. [12] further elaborated the crack coalescence and the specimen failure process under uniaxial compression using the method of combining acoustic emission technology and photographic monitoring.

Most of the existing results have studied the strength failure and crack propagation of rock materials under constant loading conditions. However, most previous studies are based on different constant rate loadings in mechanical tests on specimens, while the actual dynamic loads on rocks, such as those generated from construction, mining, structural extrusion, composite rock, earthquakes, and explosions, are not constant. The load during these two loading processes differs. There is comparatively little research on the sudden change in loading rate and its effect on the deformation strength and crack propagation of fractured rock. Therefore, based on previous research results, the rock-like material was taken as the research object in this paper and the coalesced crack was prefabricated. The fracture process of the prefabricated crack specimen was analyzed through a sudden change loading rate uniaxial loading test, combined with acoustic emission and high-definition photographic monitoring to further investigate the influence of sudden change loading rate on crack propagation evolution and failure mode of the specimen. This study can provide the theoretical basis and reference for the revelation of failure mechanism and stability assessment of rock fracture.

## 2. Test Scheme

### 2.1. Test Materials and Specimen Preparation

In order to simulate the brittle failure of the rock bridge in fractured rock mass, the rock-like model specimens were made from the model materials (consisting of white cement, sand, and

water) which were similar to rocks (brittleness and dilatancy). The ratio of the model material used was white cement:sand:water = 39:37:15 (weight ratio). The components of test mold were steel mold with an internal size length × width × height = 150 mm × 150 mm × 30 mm. In order to precast crack in the rock-like specimen, the specimen was poured into the mold and then vibrated evenly by the shaking table. After curing at room temperature for 2–2.5 h, the mica sheet with a thickness of 0.1 mm was inserted into the predetermined groove position and continued curing for about 10 h. Then, the mold was released and the mica was pulled out. After demolding the specimen, we checked the surface and flatness of the specimen, as well as the penetration of the crack. The uneven parts of specimen were polished to ensure that the flatness of the specimen could meet the test requirements. Then, the specimen was cured for 28 days.

### 2.2. Spatial Distribution of Crack

Two cracks were prefabricated and their parameters included length, angle α, and rock bridge angle β. The length of crack was 30 mm. The length of the rock bridge was 40 mm. The crack angle α was 30°. The rock bridge angle β was 60° and 90°, respectively. In order to study the change of the coalescence mode of rock bridge, the rheological propagation and coalescence process of two prefabricated cracks were researched. The spatial distribution of crack is shown in the figure. The size of the specimen is length × width × height = 150 mm× 150 mm × 30 mm. Prefabricated crack, rock bridge, prefabricated crack angle α, and the rock bridge angle β are shown in Figure 1.

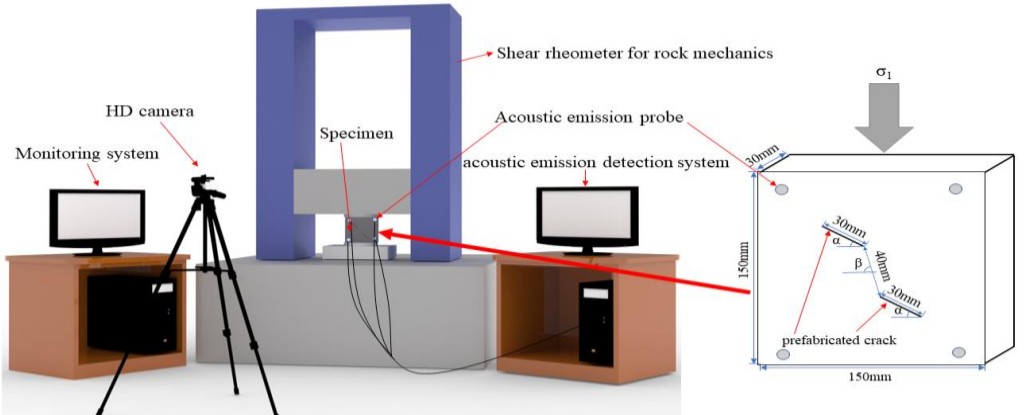

**Figure 1.** Rock mechanics test, acoustic emission system, and schematic diagram of prefabricated double fissure rock specimen.

### 2.3. Test Equipment and Test Methods

The uniaxial compression test of fractured rock-like materials was completed in the Hunan Provincial Key Laboratory of Safe Mining Techniques of Coal Mines, Hunan University of Science and Technology. The loading instrument is an RYL-600 shear rheometer produced by Changchun Chaoyang Instrument Co., Ltd. (Changchun, China), which is mainly used for shear rheological experiment of rock or concrete. It has the characteristics of large stiffness, accurate measurement, high control precision, and good stability. The loading method was controlled by displacement, with a loading rate between 0.5 and 5 mm/min. In order to reduce the end effect at the contact point of the end face of the specimen during loading and improve the uniformity of the end face compression, the end face was finely polished and coated with lubricating oil. The constant loading rate was 0.5 mm/min, 1 mm/min, 3 mm/min, and 5 mm/min, respectively.

Meanwhile, the AE equipment was used to record the AE events in the failure process of the rock-like materials. The AE equipment is PCI-2, which is produced by the PAC company. The equipment, with low noise and low price, can record the value below 17 dB in the engineering practice, with high accuracy; therefore, it is widely used in the physical tests and engineering practice.

The sudden change loading rate was divided into accelerated loading and decelerated loading, as shown in Figure 2. The acceleration stages were 0.5 mm/min to 1 mm/min, 1 mm/min to 3 mm/min, and 3 mm/min to 5 mm/min. The deceleration stages were 5 mm/min to 1 mm/min and 5 mm/min to 0.5 mm/min. The sudden change loading rate time was 50% (4.5 MPa) of the average value of the peak stress obtained by many previous tests, so that the stress–strain curve of the specimen at the sudden change loading rate time was in the linear elastic stage. The load was then continued at this rate. Due to the slow loading, high-definition video shooting and acoustic emission monitoring were conducted during the whole loading process to observe the crack initiation, coalescence, and failure of the specimen.

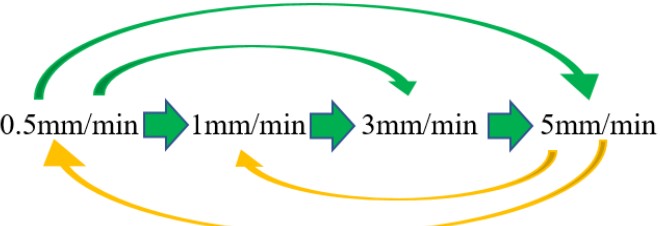

**Figure 2.** Loading rate change diagram.

## 3. Crack Evolution and Failure Characteristics of Rock Bridge Specimen at Sudden Change Loading Rate

### 3.1. Analysis of the Corresponding Relationship between Crack Propagation and Stress–Strain Curve under Uniform Loading

The crack development and propagation of rock-like specimens are reflected in the stress–strain curves. The development and generation of cracks will cause the fluctuation and change of curves [13]. Specimens #1 ($\alpha = 30°$, $\beta = 60°$) and specimen #2 ($\alpha = 30°$, $\beta = 90°$) were loaded at different rates, and then the relationship between crack propagation and stress–strain curves was analyzed under uniform loading.

The axial stress–strain relationship under uniaxial compression at different loading rates of specimen ($\alpha = 30°$, $\beta = 90°$) is shown in Figure 3. The four rates are 0.5 mm/min, 1 min/min, 3 mm/min, and 5 mm/min, respectively. Table 1 lists the peak strength $q_c$ and maximum strain $\varepsilon_c$ (that is, the strain corresponding to the peak axial stress) of the specimen at the corresponding loading rate.

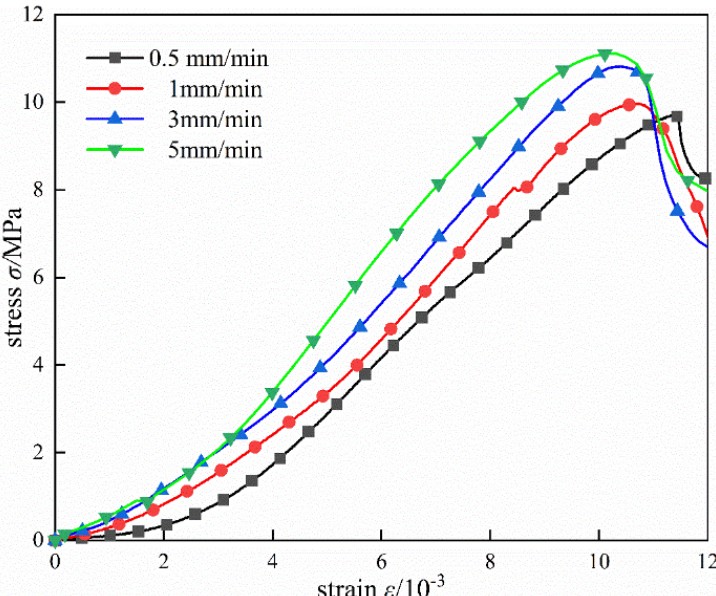

**Figure 3.** The stress–strain relationship of rock−like under different loading rates.

**Table 1.** Peak Strength, Strain, and Elastic Modulus of Specimens under Uniaxial Condition.

| Loading Rate $v$/(mm/min) | Peak Strength $q_c$/mpa | Peak Strain $\varepsilon_c$/$10^{-3}$ | Elastic Modulus/GPa |
|---|---|---|---|
| 0.5 | 9.70 | 11.37 | 1.20 |
| 1 | 9.96 | 10.70 | 1.41 |
| 3 | 10.81 | 10.39 | 1.42 |
| 5 | 11.11 | 10.30 | 1.79 |

It can be seen from the table that, with the increase in loading rate, the peak strength and elastic modulus increase and the maximum strain decreases [4]. The test results indicate that the stress–strain curve changes with the loading rate. The rock specimen goes through the stages of pore fracture compaction, elastic deformation, stable micro elastic fracture development, unstable fracture development, and post-fracture [14].

The general behavior of crack evolution of specimens under uniform loading is as follows: the stress–strain curve of specimens varies with the loading rate. When the loading rate increases, the elastic modulus increases, the peak strength also increases, and the peak strain decreases. During loading, as shown in Figure 4, the remote cracks usually appear first, and then the partial cracks of the rock bridge begin to develop. With the increase of loading rate, the crack propagation at the precast crack tip changes gradually from tensile crack to shear crack. The higher the loading rate, the less the macroscopic cracks of the specimen. At low rate, the internal defects of the specimen have enough time to develop and expand. For example, when loading at low rate, the rock bridge crack is a primary crack, and when loading at high rate, the rock bridge crack is a double crack. This is because microcracks in the rock bridge region fully develop at low rate; then, two precast cracks continuously develop in the stress concentration area and connect through the microcracks, resulting in a primary crack. When loading at high rate, there is not enough microcrack initiation in the rock bridge area, and the cracks of both sides develop relatively independently, leading to double cracks in the rock bridge area.

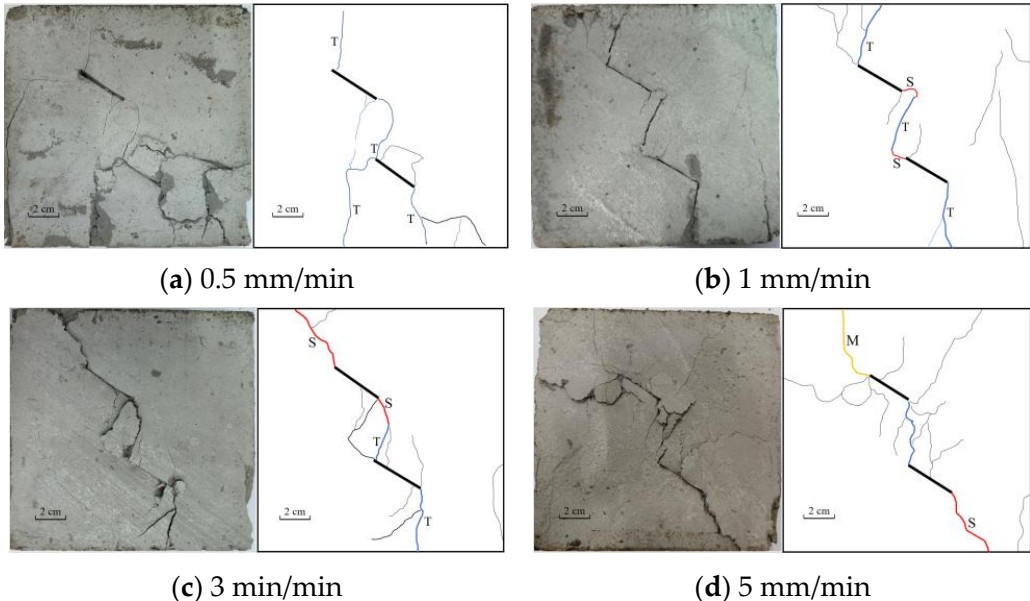

(**a**) 0.5 mm/min

(**b**) 1 mm/min

(**c**) 3 min/min

(**d**) 5 mm/min

**Figure 4.** Crack development diagram of crack specimen under uniform loading.

The ultimate failure modes of specimens under different loading rates include shear failure, tensile failure, and tensile–shear mixed failure.

As can be seen from Figure 5, tensile failure mode is the main choice of low-rate loading specimens. The change of loading speed will lead to the change of failure mode. For example, the specimens with a loading rate of 3 min/min are tensile–shear mixed

failures; when loading rate is 5 mm/min, the ultimate failure mode of the specimens is mainly shear failure.

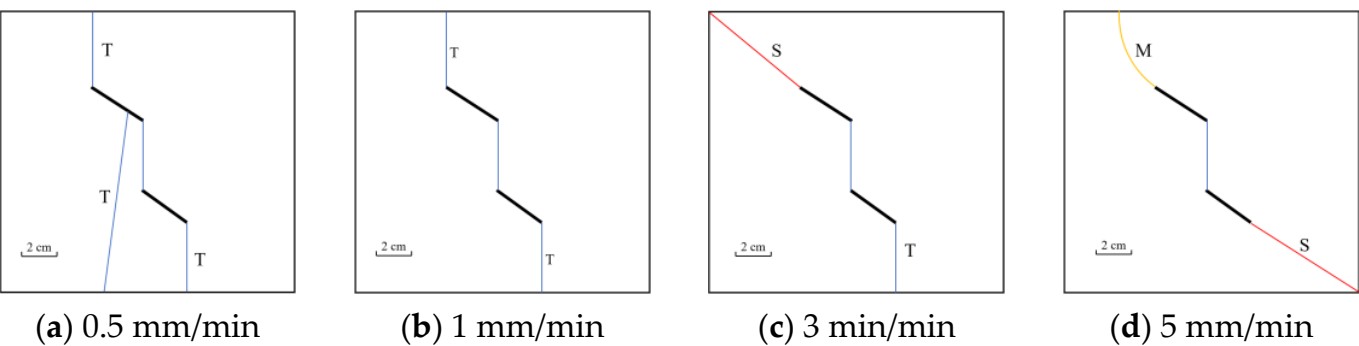

**(a)** 0.5 mm/min     **(b)** 1 mm/min     **(c)** 3 min/min     **(d)** 5 mm/min

**Figure 5.** Crack propagation mode diagram of crack test under uniform loading.

Figure 6 shows the crack propagation process of specimen #1 under uniaxial compression.

**(a)** 0.5 mm/min              **(b)** 1 mm/min

**(c)** 3 mm/min              **(d)** 5 mm/min

**Figure 6.** Correspondence relationship between crack propagation and stress–strain curve of typical specimen.

*3.2. Analysis of Crack Propagation and Failure Modes of Specimens under Sudden Change Loading Rate Loading*

Under sudden change loading rate loading, the stress–strain relationship of the sample changes with loading rate. The specimen accumulates energy continuously at the initial stage of loading, and produces new cracks to release energy after exceeding a certain critical state. The final failure mode of the specimen often depends on the type of crack. There are two main different cracks during the process of uniaxial failure: tensile crack and shear crack. According to the crack types classified by Zhao Yanlin and Cao Ping, et al. [15–22], secondary shear crack and wing tensile crack are the main factors leading to the instability failure of specimens under uniaxial compression. The cracks observed in this test were wing crack, tensile resistance crack, remote crack, transverse crack, secondary coplanar crack, and secondary noncoplanar crack, as shown in the Figure 7. The loading test results are explained in Tables 2 and 3. Different specimen numbers represent different transmission modes. For example, the number 0.5–1 indicates that the specimen changes from 0.5 mm/min to 1 mm/min. Crack tracks were drawn by hand. The main tensile cracks are pictured by blue lines. The shear cracks are drawn by red lines in the sketch. According to the mechanism and track of crack initiation, the failure modes are divided into different ones.

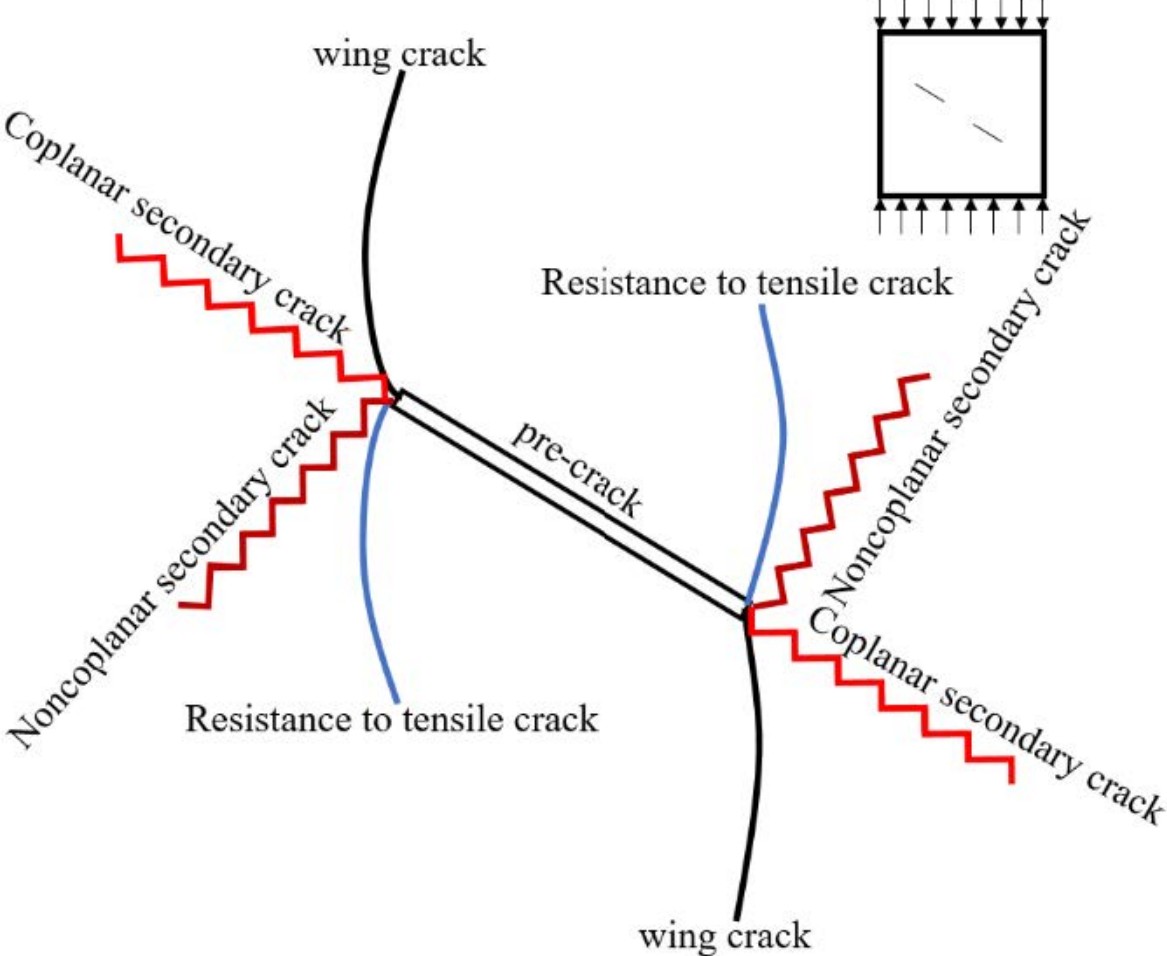

**Figure 7.** Crack initiation type diagram.

**Table 2.** Specimens #1 of different sudden change rate crack propagation diagram.

| Specimen Number | Realistic Graph | Sketch | Failure Mode Diagram |
| --- | --- | --- | --- |
| 0.5-1 |  |  |  |
| 0.5-3 |  |  |  |
| 0.5-5 |  |  |  |
| 1-3 |  |  |  |
| 3-5 |  |  |  |
| 5-0.5 |  |  |  |
| 5-1 |  |  |  |

**Table 3.** Specimens #2 of different sudden change rate crack propagation diagram.

| Specimen Number | Realistic Graph | Sketch | Failure Mode Diagram |
| --- | --- | --- | --- |
| 0.5-1 |  |  |  |
| 0.5-3 |  |  |  |
| 0.5-5 |  |  |  |
| 1-3 |  |  |  |
| 3-5 |  |  |  |
| 5-0.5 |  |  |  |
| 5-1 |  |  |  |

Crack propagation of specimen #1 at different loading rates:

A wide tensile wing crack was developed at the left tip of the two prefabricated cracks in the 0.5-1 specimen at the same time. The expansion direction of the crack was approximately perpendicular to the prefabricated crack, then gradually deviated towards the axial stress direction, and finally expanded towards the specimen end along the axial stress direction. A downward shear crack was generated at the right tip of the upper prefabricated crack, which intersected with the secondary coplanar crack generated at the left tip of the lower prefabricated crack. The rock bridge was connected, resulting in structural failure and instability of the specimen.

The remote crack appeared first in specimen 0.5-3. Tensile wing cracks were generated downward and upward at the right tip and left tip of the upper and lower prefabricated cracks, respectively, until the crack expanded to the end; the specimen was completely destroyed and the rock bridge was not connected.

The remote crack of specimen 0.5-5 first appeared. The right tip of the upper prefabricated crack showed an inclined upward tension crack to the left and a downward tension wing crack, then gradually tilted toward the axial stress direction. With the passage of time, the upper prefabricated crack was compacted. Two prefabricated cracks expanded secondary cracks to make the rock bridge connected until the strength failure of the specimen occurred.

The right tip and left tip of the upper and lower prefabricated cracks of the specimen 1-3 produced downward and upward tensile wing cracks, respectively, to connect the rock bridge, and continued loading. The left tip of the upper prefabricated crack and the right tip of the lower prefabricated crack expanded shear cracks along the parallel direction of the prefabricated crack, respectively, until the crack was connected and the strength of the specimen failed.

For specimens 3-5, the remote crack appeared first. The stress perpendicular to the axial direction was produced in the middle of the lower prefabricated crack extended toward the upper end. The tensile crack perpendicular to the prefabricated crack was generated at the left tip of the lower prefabricated crack. The shear crack occurred at the right tip of the upper prefabricated crack to connect the rock bridge, and the shear crack occurred at the left tip of the upper prefabricated crack.

The remote crack of the 5-0.5 specimen appeared. The upper right corner of the specimen spalled off. Tensile wing cracks came out at the tips of two prefabricated cracks. The crack width increased continuously until the specimen was completely destroyed; the rock bridge was not connected.

A crack perpendicular to the prefabricated crack expanded at the left tip of the upper part of specimen 5-1. Then, it gradually evolved into transverse crack. The shear crack propagated at the right tip of the upper prefabricated crack and the lower prefabricated crack until crack was connected, and the specimen was structurally damaged.

Crack propagation of specimen #2 at different loading rates:

The upper prefabricated crack and the lower prefabricated crack of the 0.5-1 specimen spread upward and downward tensile wing cracks, respectively, thus connecting the rock bridge. With the continuous loading, the two prefabricated cracks both spread out tensile wing cracks, and the specimen was connected instantaneously. The specimen was completely destroyed accompanied by a bursting sound.

The whole process of specimen 0.5-1 was similar to that of specimen 0.5-1. The remote cracks appeared, then two prefabricated cracks expanded the tensile wing cracks independently until the rock bridge was connected and the specimen was damaged by tensile.

The remote crack of specimen 0.5-5 first appeared, and the rock bridge crack developed. Two prefabricated cracks were compacted. Shear cracks and tensile wing cracks propagated on the left tip of the upper prefabricated crack and the right tip of the lower prefabricated crack, accompanied by a cracking sound. The specimens showed structural failure and instability, presenting mixed failure of tensile shear.

The rock bridge crack developed first in specimens 1-3, and the shear crack expanded at the left tip of the upper prefabricated crack and continued to load with surface spalling. At the right tip of the lower prefabricated crack, the tensile wing crack extended and connected instantaneously. The specimen was damaged and unstable.

The remote crack of specimens 3-5 developed first. Shear crack and tensile wing crack propagated simultaneously at the right tip of the upper prefabricated crack to connect the rock bridge. The shear crack extended to the left tip of the upper prefabricated crack, and the tensile crack extended to the right tip of the lower prefabricated crack. With the splitting sound, the crack was connected and the specimen was destroyed.

The whole process of specimen 5-0.5 was similar to that of specimen 0.5-1. The rock bridge crack developed first, and the tensile wing crack expanded from the upper prefabricated crack and the lower prefabricated crack, respectively, resulting in tensile failure of the specimen.

The remote crack of specimen 5-1 was first generated. The rock bridge expanded sawtooth crack. The upper left corner of the upper prefabricated crack expanded the tensile shear crack. A crack parallel to the prefabricated crack generated at the right tip of the lower prefabricated crack, gradually deflecting toward the axial stress direction. As the loading continued, the specimen cracks coalesced and gave off a cracking sound, resulting in structural damage and instability of the specimen.

According to the experimental results, the statistical results of the ultimate failure mode of the rock specimens under the influence of different sudden change loading rates are shown in Tables 4 and 5. 'iv' represents the initial velocity and 'fv' represents the final velocity. "T" represents the tensile failure of the primary crack. "S" denotes the shear failure of the primary crack. "M" expresses the tensile shear mixed crack of the primary crack. The general pattern is sketched in Figure 8. The failure mode is closely related to the rate change in the loading process. When the initial loading rate and sudden change loading rate are both low, the failure mode of the specimen is tensile failure. When the initial rate and sudden change loading rate are both high, the failure mode tends to be shear failure. The mode between the two ends is tensile–shear mixed failure. It is worth noting that there is no accurate and obvious dividing line of failure modes in the test, but a transition evolution between different modes.

**Table 4.** Statistical table of 30–60 failure modes of specimens.

| Rate of Loading | fv 0.5 mm/min | fv 1 mm/min | fv 3 mm/min | fv 5 mm/min |
|---|---|---|---|---|
| iv 0.5 mm/min | T | M | T | S |
| iv 1 mm/min | \ | M | M | \ |
| iv 3 mm/min | \ | \ | M | M |
| iv 5 mm/min | T | S | \ | S |

**Table 5.** Statistical table of 30–90 failure modes of specimens.

| Rate of Loading | fv 0.5 mm/min | fv 1 mm/min | fv 3 mm/min | fv 5 mm/min |
|---|---|---|---|---|
| iv 0.5 mm/min | T | T | T | S |
| iv 1 mm/min | \ | T | M | \ |
| iv 3 mm/min | \ | \ | M | M |
| iv 5 mm/min | T | T | \ | S |

The general characteristics of the crack evolution of the specimen with sudden change loading rate are as follows: the remote crack usually appears at the upper and lower ends first, followed by the tensile crack. When the specimen is damaged in the form of tensile, it will be accompanied by brittle sound. When shear failure is predominant, the breakage degree of the specimen is serious. During the uniaxial compression test, when the loading rate changes suddenly, the stress–strain curve of the complete specimen is no longer maintained on the stress–strain curve which is corresponding to the constant loading

rate, but shows the deformation characteristics of high rigidity and approximate elasticity. The stress–strain curve after loading rate mutation gradually tends to the stress–strain curve corresponding to the constant rate loading at the mutated rate. In the loading process, the failure mode of the specimen is mainly shear failure when the rate is higher. If the rate is lower, the failure mode of the specimen tends to be tensile failure, which indicates that the final failure mode of the specimen mainly depends on the loading rate of the rock during the crack propagation stage. The loading rate has little effect on the final failure form of rock in the compaction stage.

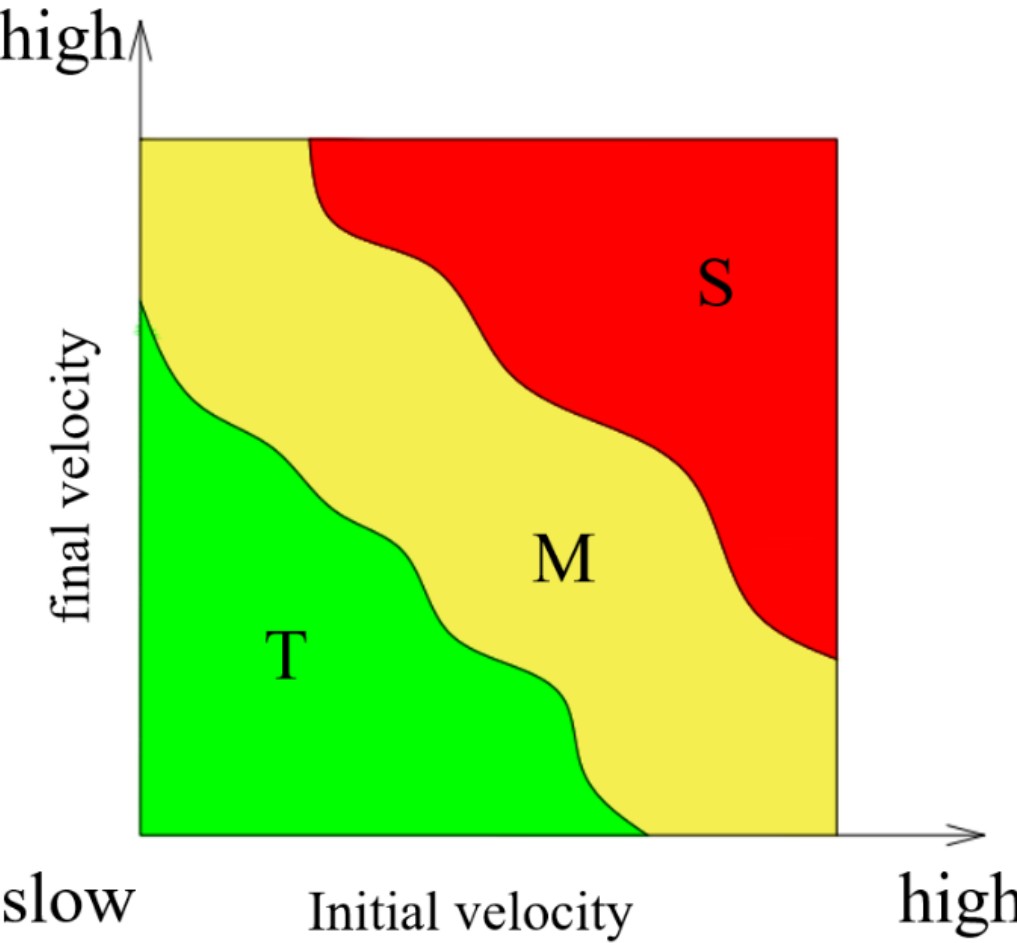

**Figure 8.** Diagram of relationship between sudden change rate loading mode and final failure mode.

*3.3. Correspondence of AE Information under Sudden Change Loading Rate*

In the process of crack propagation and failure of rocks, the rapid release of internal energy will generate transient elastic waves which can be collected by acoustic emission monitoring system. An 8-channel acoustic emission detection system (Model PCI-2) was used to record the acoustic emission signals of the specimens. The specimens under sudden change loading rate were analyzed with the help of the time–stress curve, crack propagation video, and three-dimensional acoustic emission positioning diagram.

Due to paper length limitation, the following figure only shows the AE spatio response diagram of specimen #2. Combined with Figure 9, the whole loading process can be successively divided into several stages according to the AE evolution characteristics: ① compaction stage, ② linear elastic stage, ③ stable crack propagation stage, ④ unstable crack propagation stage, ⑤ post-peak stage. In the loading process, the time–stress AE curves of accelerated loading and decelerated loading are significantly different because of the loading differences. For specific characteristics, see Figure 9.

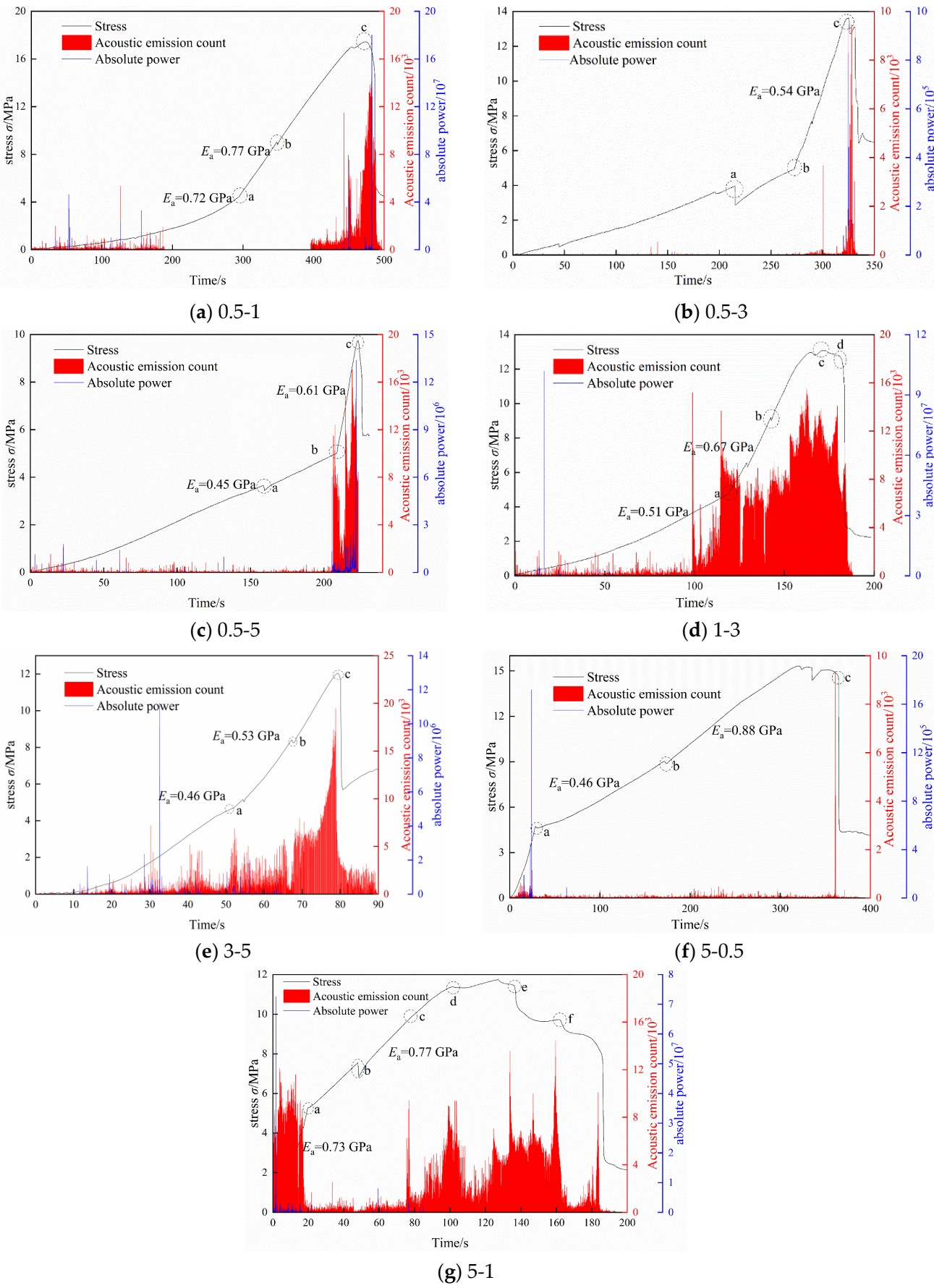

**Figure 9.** Acoustic emission characteristics of fractured rock samples.

Small-span accelerated loading, such as in the 0.5-1 specimen: ① compaction stage, a small amount of acoustic emission signals is generated, amplitude distribution is in 30~45 dB, and absolute energy is less than $2 \times 10^6$ aJ. ② Linear elastic stage, AE signals are inactive, amplitudes mainly distribute in the range of 30~50 dB, occasionally 69 dB AE, and the absolute energy is mostly less than $1 \times 10^6$ aJ. ③ Stable crack propagation stage, acoustic emission signals are fast and active, and accumulated energy increases rapidly. Amplitude distribution is between 50 and 80 dB, and amplitude is 97 dB in the middle stage. ④ Unstable crack propagation stage, the signal surges, and the accumulated energy increases rapidly, with the amplitude distribution between 70 and 80 dB. The absolute energy is mostly less than $2 \times 10^8$ aJ. ⑤ Post-peak stage, when the stress drops, the amplitude of acoustic emission signal decreases immediately within 30–40 dB, and the absolute energy is less than $1 \times 10^6$ aJ.

Large-span accelerated loading, such as in 0.5-5 specimen: ① compaction stage, signal values appear, amplitude distribution is between 30 and 50 dB, and absolute energy is mostly less than $1 \times 10^6$ aJ. ② Linear elastic stage, a relatively high acoustic emission signal peak appears, and the amplitude is distributed in the range of 50~80 Db. The acoustic emission of 96 dB appears in the sudden change loading rate, and the absolute energy is mostly less than $1 \times 10^7$ aJ. ③ Stable crack propagation stage, and the AE signal responds with high value and gradually increases, with the amplitude ranging from 65 to 90 dB. The absolute energy is mostly less than $1 \times 10^8$ aJ. ④ Unstable crack propagation stage, the stress reaches the peak, the amplitude is between 80 and 100 dB, and the absolute energy is less than $1 \times 10^8$ aJ. ⑤ Post-peak stage, the stress drops precipitously, with the amplitude of 30–50 dB, and the absolute energy is less than $1 \times 10^6$ aJ.

Deceleration loading, such as in 5-1 specimen: ① compaction stage, loading rate is high; acoustic emission signal is obvious, amplitude distribution is 60~85 dB, and absolute energy is less than $1 \times 10^8$ aJ. ② Linear elastic stage, the AE signal is always at a low level, the amplitude is between 30 and 45 dB, and the absolute energy is mostly less than $1 \times 10^6$ aJ. ③ Stable crack propagation stage, the acoustic emission signals are dense, and the response begins to be dense. The amplitude distribution is between 50 and 70 dB, and the absolute energy is mostly less than $1 \times 10^7$ aJ. ④ Unstable crack propagation stage, the acoustic emission detection signal surges, with the amplitude between 70 and 90 dB, and the absolute energy mostly less than $1 \times 10^8$ aJ. ⑤ Post-peak stage; the AE signal drops, the amplitude distribution is 30~50 dB, and the absolute energy is less than $1 \times 10^5$ aJ.

Combined with Figure 10 and Table 3, it can be seen that the deformation modulus at the linear elastic stage of the specimen is roughly the same. The final failure mode is the same, even if the rate before the change is different and the rate after the change is the same.

At low loading rate stage, AE count is generally low, while at high loading rate stage, AE count is generally high.

From high rate to low rate, the AE counting image presents a "U" shape or a "W" shape. This is because the AE signal is at a high value in the early high-rate loading. When the crack development stage reaches, the rapid crack development will also lead to the rapid increase of AE signal. When the specimen is about to be destroyed, the AE signal will also respond with a high value until the specimen is destroyed and then the signal decreases and disappears.

Based on the above analysis, the crack propagation mode and failure mode are obviously affected by the loading rate.

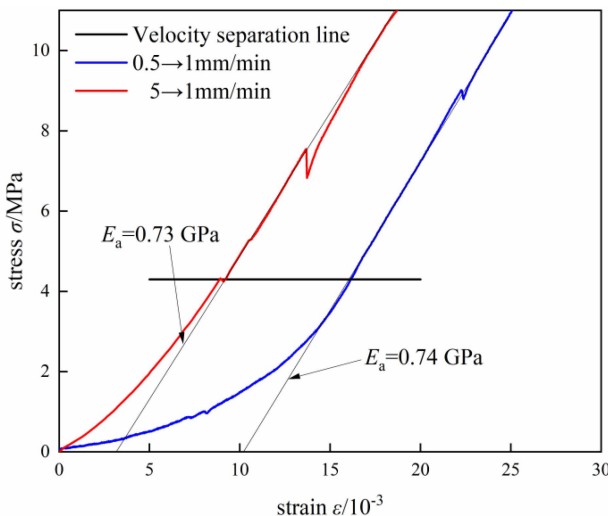

**Figure 10.** Comparison diagram of stress–strain in elastic stage of velocity line after same sudden change speed.

## 4. Discussion on the Mechanism of Crack Evolution Influenced by Sudden Change Loading Rate

In this section, the crack propagation process of prefabricated crack specimens under different sudden change loading rate is comprehensively analyzed by means of full stress–strain curve, crack propagation diagram, and acoustic emission 3D positioning (Figures 11 and 12).

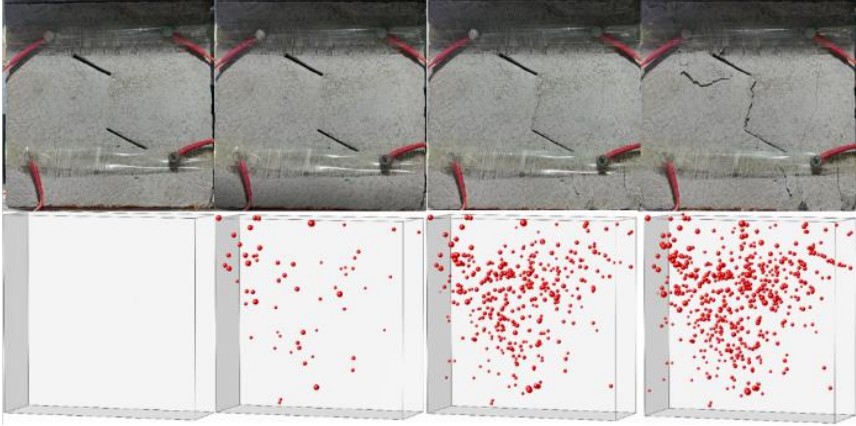

**Figure 11.** 90-5-1 specimen surface macrocrack and acoustic emission crack location information map.

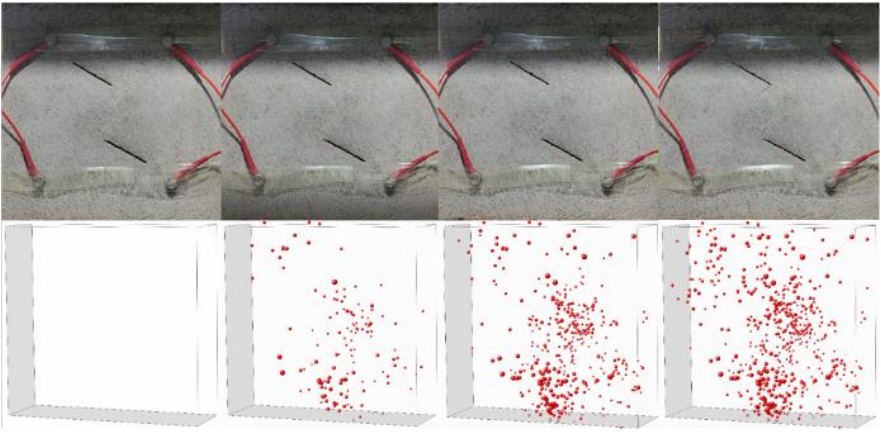

**Figure 12.** 90-1-3 specimen surface macrocrack and acoustic emission crack location information map.

There is an obvious sudden change loading rate effect on the fracture evolution of the prefabricated double-crack, rock-like specimens, and there is an obvious turning point in the large-span sudden change-rate loading. The sudden change-rate loading effect of crack propagation is revealed by analyzing the difference between the small-span sudden change-rate loading and the large-span sudden change-rate loading.

When the loading rate is small (<3 mm/min), the AE curve is inactive, which is mainly due to the need of energy for crack development and propagation. The higher the loading rate before rate change is, the more active the AE signal is, and the greater the change of the internal stress of the unit specimen is. Under the condition of high disturbance and rapid accumulation of energy, the crack is more likely to germinate and expand, releasing larger energy, and the higher the AE count is.

When the loading rate is small, the final failure mode of the 60° specimen is the tensile failure formed along the two ends of the prefabricated crack. When the loading rate is large, the final failure mode of the specimen is the connected rock bridge between the two prefabricated cracks. The cracks at both ends of the prefabricated crack turn into the tensile–shear failure mode or the shear failure mode.

From low rate to high rate, stress drop may be induced, resulting in crack generation and a peak in the image of acoustic emission counting. Combined with Wang Xiaoran's theory of solid fracture nucleation of unstable fractured sandstone [16], when stress rises, the formation rate of microfractures is exponentially related to stress. A small stress increment will produce a large number of defects. Any small stress increment may lead to the cascade development of microfractures. When the microfracture density exceeds the critical threshold, it will lead to the fracture of locked solid. Therefore, the local stress drop will occur due to the instantaneous acceleration at the time of rate change.

## 5. Conclusions

(1) There are obvious differences in the mechanical properties and failure modes of specimens under different constant loading rates. The higher the loading rate is, the higher the peak strength and elastic modulus are, and the smaller the corresponding peak strain is. With the increase of loading rate, the crack initiation type changes from tensile wing crack to shear crack. The rock bridge area changes from smooth tensile crack to sawtooth shear crack.

(2) During the process of sudden change-rate loading, the specimen exhibits an approximate elastic deformation. The mutational stress–strain curve is similar to the corresponding stress–strain curve under constant loading rate.

(3) Combined with acoustic emission detection, it is found that when the rates before mutation have no differences, the stress–strain curves of the specimens were similar with the same rates after mutation.

At the low-rate loading stage, AE count is generally low. While at the high-rate loading stage, AE count is generally high.

The change from low-rate loading to high-rate loading easily induces stress drop, resulting in crack generation and an increase in AE count. Therefore, the potential surge of loading rate can be inferred by the stress drop and acoustic emission surge in engineering monitoring applications.

From high-rate loading to low-rate loading, the AE counting image presents a "U" shape or a "W" shape. This is because the AE signal was at a high value in the early high-rate loading. When the crack development stage arrives, the rapid crack development could also lead to the rapid increase of AE signal. When the specimen is damaged, the AE signal suddenly increases and then disappears.

(4) The failure mode of the specimen is determined by the rate of the specimen in the crack development period. In the early period, a certain range of low-rate loading has little influence on the specimen, as shown in Figure 10. When the early-stage stress is low, the AE count of the specimen is high, which will cause certain damage inside the specimen. As a result, the final peak stress and failure mode may be different even if the rate is the same after changing the rate.

**Author Contributions:** Methodology, Y.Z.; Writing—original draft, X.H.; Writing—review & editing, M.W. and J.L.; Supervision, W.C.; Funding acquisition, W.W. All authors have read and agreed to the published version of the manuscript.

**Funding:** This research was financially supported by the National Natural Science Foundation of China (No. 52274194), and Hunan Provincial Natural Science Foundation of China (No. 2021JJ30265).

**Conflicts of Interest:** The authors declare that they have no competing interest.

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
