# Peer review of "Study on Crack Propagation of Rock Bridge in Rock-like Material with Fractures under Compression Loading with Sudden Change Rate"

_applsci, doi:10.3390/app13074354_

Round 1

Reviewer 1 Report

The aim of this research is to explore how the rate of sudden loading change affects the failure of rock mass specimens. The study includes some value observations and discussions for the readers. Based on the content, the reviewer believes that this paper can be accepted for publication after considering some points.

1.    It is recommended to carefully check the spelling and grammar mistakes. For example, before putting a reference in page 1, we need to leave a single space. This also applies to other pages.

2.    How did the authors properly distinguish between different types of cracks in their experimental tests? What was criteria used for this determination?

3.    In page 1: “In the process of sudden change loading, the specimen shows the characteristics of approximate elastic deformation, and the stress-strain curve after sudden loading is similar to the corresponding stress-strain curve at the corresponding constant loading rate.” What results showing this conclusion?

Author Response

Responses to review comments

Comments and Suggestions for Authors

The aim of this research is to explore how the rate of sudden loading change affects the failure of rock mass specimens. The study includes some value observations and discussions for the readers. Based on the content, the reviewer believes that this paper can be accepted for publication after considering some points.

Response: Thanks for your comments, your suggestions are conducive to improving the quality of our manuscript. According to your comments, questions were answered, some modifications were made, and the changes were highlighted by red color, which can be seen in the revised manuscript.

  1. It is recommended to carefully check the spelling and grammar mistakes. For example, before putting a reference in page 1, we need to leave a single space. This also applies to other pages.

Response: Thanks for your comments. According to your suggestions, our manuscript was revised and polished by the American Journal Experts, the editing proof was added as an attachment, and the changes were highlighted by red color.

  1. How did the authors properly distinguish between different types of cracks in their experimental tests? What was criteria used for this determination?

Response: Thanks for your comments. The classification of different types of cracks is depends on the stress occurred on crack surfaces or the shape of new generated cracks, which can be seen in Fig. 7, and the corresponding statements can be seen in the reference [1]. I hope our answer can meet your satisfactory.

  1. In page 1: “In the process of sudden change loading, the specimen shows the characteristics of approximate elastic deformation, and the stress-strain curve after sudden loading is similar to the corresponding stress-strain curve at the corresponding constant loading rate.” What results showing this conclusion?

Response: Thanks for your comments. In our experiments, when sudden change loading occurred, different deformation ratio would appeared, and the strain ratio is linear in most cases, which can be observed in Fig. 10 . I hope my answer can meet your satisfactory.

Reviewer 2 Report

The article deals with the experimental observation of coupled cracks on rock-like material subjected to changes in strain rate. There are several issues to be addressed in the paper.

I would add “compression loading” somewhere in the title. The direction of the applied load significantly changes the crack propagation.

The loading scheme described in page 3 does not match figure 2. How were those rates picked?

Fig 3. Shows the used rock-like material is likely brittle and that it can store more energy at higher strain rates. A microscopy analysis may shed light on the deformation mechanism to confirm the structure-properties relation.

Please include details of the AE equipment.

A general English correction is needed; there are several missing articles, commas, and prepositions.

In section 2.1, was the crack a through-crack ?

I suggest the authors combine Fig 2 and table 1 in a force versus time plot to better comprehend the loading scheme

Fig 4, 5, and 6 add scale to understand the magnitude of crack kinking. Unfortunately, the size of the pics (and the PDF format) does not let me appreciate them well.

Please invert Fig 7 to coincide the schematics with the angle of the precrack

Fig 8 should not be qualitative. The way it is, it does not give much information.

From Fig 3, I see the Elastic modulus is not heavily dependent on strain rate. The slopes are almost parallel, and their difference is about 10%, whereas the difference in load is about 10 times.

On page 6 it reads: "The increase of loading rate will make failure mode change from tensile to shear" So why the change? Perhaps the compression at a higher rate overcomes the friction and crack rugosity that prevents shear from happening? The authors may want to examine how the stress field develops ahead of the crack tip. See  Yates 2008 where they showed how coupled cracks propagate towards or away from each other depending on the stress field. See Vormwald and Yang (2017), where they explained the tensile to shear change due to the change on mode-mixity ratio or Diaz (2021) where they tested several models for monotonic crack kinking propagation.

Suggested references:

Yates 2008. Crack paths under mixed mode loading. doi:10.1016/j.engfracmech.2007.05.014

Vormwald and Yang (2017)  Fatigue crack growth simulation under cyclic non-proportional mixed mode loading. DOI: 10.1016/j.ijfatigue.2017.04.014

Diaz and Freire (2021)  LEFM crack path models evaluation under proportional and non-proportional load in low carbon steels using digital image correlation data. DOI 10.1016/j.ijfatigue.2021.106687

Author Response

Responses to review comments

Comments and Suggestions for Authors

The article deals with the experimental observation of coupled cracks on rock-like material subjected to changes in strain rate. There are several issues to be addressed in the paper.

 Response: Thanks for your comments, your suggestions are helpful for improving our manuscript quality.

I would add “compression loading” somewhere in the title. The direction of the applied load significantly changes the crack propagation.

Response: Thanks for your comments. According to your suggestions, the “compression loading” was added in the title, which can be seen in the revised manuscript, and the changes were highlighted by red color.

The loading scheme described in page 3 does not match figure 2. How were those rates picked?

Response: Thanks for your comments. The experimental process is described in Part 2.3 (Test equipment and test methods), in our tests, The sudden change loading rate was divided into accelerated loading and decelerated loading, as shown in Figure 2. The acceleration stages were 0.5mm/min to 1mm/min, 1mm/min to 3mm/min, 3mm/min to 5mm/min. The deceleration stages were 5mm/min to 1mm/min and 5mm/min to 0.5mm/min, which can be seen in Figure 2. Those rates are picked based on our experience. I hope our answer can meet your satisfactory.

Fig 3. Shows the used rock-like material is likely brittle and that it can store more energy at higher strain rates. A microscopy analysis may shed light on the deformation mechanism to confirm the structure-properties relation.

Response: Thanks for your comments. The rock-like materials is brittle, it can be broken when the peak strength is reached in a short time. As a matter of fact, the failure process of the rock-like materials with preexisting cracks were discussed and analyzed in detail, and the deformation analysis is included in the failure process, I hope our answer can meet your satisfactory.

Please include details of the AE equipment.

Response: Thanks for your comments. Some details information about AE equipment was added in the revise manuscript, which is highlighted in red color.

A general English correction is needed; there are several missing articles, commas, and prepositions.

Response: Thanks for your comments, your suggestions are conducive to improving the quality of our manuscript. According to your comments, questions were answered, some modifications were made, and the changes were highlighted by red color, which can be seen in the revised manuscript.

In section 2.1, was the crack a through-crack ?

Response: Thanks for your comments. The fabricate crack is through-crack, which means, the length of fabricate crack is 30 mm, which is the same as the thickness of the specimen. And the fabricate crack is across the specimen vertically.

I suggest the authors combine Fig 2 and table 1 in a force versus time plot to better comprehend the loading scheme

Response: Thanks for your comments. In fact, the content of Fig 2 is quit different from Table 1. Fig 2 is about the process of physical tests, however, Table 1 are some basic parameters of rock-like materials, thereof, Fig 2 and Table 1 should not be combined.

Fig 4, 5, and 6 add scale to understand the magnitude of crack kinking. Unfortunately, the size of the pics (and the PDF format) does not let me appreciate them well.

Response: Thanks for your comments, the ticks of coordinate in Fig 4, 5 and 6 were added, which can be seen in the revised manuscript.

Please invert Fig 7 to coincide the schematics with the angle of the precrack

Response: Thanks for your comments, Fig. 7 was revised according to your suggestions, which can be seen in the revised manuscript.

Fig 8 should not be qualitative. The way it is, it does not give much information.

Response: Thanks for your comments. Fig.8 should be retained, it is because that the relationship between sudden change loading rate and final failure mode can be observed from this figure, which is helpful for understanding the manuscript.

From Fig 3, I see the Elastic modulus is not heavily dependent on strain rate. The slopes are almost parallel, and their difference is about 10%, whereas the difference in load is about 10 times.

Response: Thanks for your comments. The modulus is independent on loading rate, from Fig. 3, it can be observed that the incline ratios are almost the same, these lines are almost parallel, the maximum error of Young’s modulus is some 10%, however, the difference among the peak strength is about 10 times.

On page 6 it reads: "The increase of loading rate will make failure mode change from tensile to shear" So why the change? Perhaps the compression at a higher rate overcomes the friction and crack rugosity that prevents shear from happening? The authors may want to examine how the stress field develops ahead of the crack tip. See  Yates 2008 where they showed how coupled cracks propagate towards or away from each other depending on the stress field. See Vormwald and Yang (2017), where they explained the tensile to shear change due to the change on mode-mixity ratio or Diaz (2021) where they tested several models for monotonic crack kinking propagation.

Yates 2008. Crack paths under mixed mode loading. doi:10.1016/j.engfracmech.2007.05.014

Vormwald and Yang (2017)  Fatigue crack growth simulation under cyclic non-proportional mixed mode loading. DOI: 10.1016/j.ijfatigue.2017.04.014

Diaz and Freire (2021)  LEFM crack path models evaluation under proportional and non-proportional load in low carbon steels using digital image correlation data. DOI 10.1016/j.ijfatigue.2021.106687

Response: Response: Thanks for your comments. From analysis of the physical tests, it can be concluded that the increase of loading rate will make failure mode change from tensile to shear. However, the theoretical analysis was not made in our manuscript, and it would be our next tasks.

References:

[1] ZHAO Yanlin; WAN Wen; WANG Weijun; ZHAO Fujun; CAO Ping Compressive-shear rheological fracture of rock-like cracks and subcritical crack propagation test and fracture mechanism. Chinese Journal of Geotechnical Engineering 2012, 34, 1050–1059.

Round 2

Reviewer 2 Report

The authors have partially addressed the observations made on the 1st round.

The loading scheme described in page 3 does not match figure 2."The sudden change loading rate was divided into accelerated loading and decelerated loading, as shown in Figure 2. The acceleration stages were 0.5mm/min to 1mm/min, 1mm/min to 3mm/min, 3mm/min to 5mm/min. The deceleration stages were 5mm/min to 1mm/min and 5mm/min to 0.5mm/min, which can be seen in Figure 2." Figure 2 only shows 0.5mm/min to 3mm/min and 0.5mm/min to 5mm/min.

Fig 4, 5, and 6 add scale to understand the magnitude of crack kinking. Ticks were added but the size does not permit appreciate them well. I hope the editorial office picks on that.   

Fig 8 should not be qualitative. There is no way to tell when failure goes from Tensile to Shear. The axis ought to have a scale. Otherwise, Fig 8 is merely illustrative. There is no way to tell how much is high or slow.

I still think that a quantitative analysis of the stress field may shed a deeper insight on how the crack goes from Tensile to shear.

Suggested references (where the crack path has been adressed quatitatively):

Yates 2008. Crack paths under mixed mode loading. doi:10.1016/j.engfracmech.2007.05.014

Vormwald and Yang (2017)  Fatigue crack growth simulation under cyclic non-proportional mixed mode loading. DOI: 10.1016/j.ijfatigue.2017.04.014

Diaz and Freire (2021)  LEFM crack path models evaluation under proportional and non-proportional load in low carbon steels using digital image correlation data. DOI 10.1016/j.ijfatigue.2021.106687

Author Response

Responses to review comments

The authors have partially addressed the observations made on the 1st round.

The loading scheme described in page 3 does not match figure 2."The sudden change loading rate was divided into accelerated loading and decelerated loading, as shown in Figure 2. The acceleration stages were 0.5mm/min to 1mm/min, 1mm/min to 3mm/min, 3mm/min to 5mm/min. The deceleration stages were 5mm/min to 1mm/min and 5mm/min to 0.5mm/min, which can be seen in Figure 2." Figure 2 only shows 0.5mm/min to 3mm/min and 0.5mm/min to 5mm/min.

Reply: Thanks for your comment. The experimental process is described in Section 2.3 (Testing Equipment and Test Methods), in our test, the mutation loading rate is divided into accelerated loading and decelerated loading, as shown in Fig. 2. The acceleration stages are 0.5mm/min to 1mm/min, 1mm/min to 3mm/min, 3mm/min to 5mm/min. The deceleration stages are 5 mm/min to 1 mm/min and 5 mm/min to 0.5 mm/min, respectively. For clarity, we have reworked the figure by bolding the arrows between the numbers, changing the accelerated loading indicator arrows to green, and the decelerating loading indicator arrows to yellow. Hope our answer can satisfy you.

Fig 4, 5, and 6 add scale to understand the magnitude of crack kinking. Ticks were added but the size does not permit appreciate them well. I hope the editorial office picks on that.   

Reply: Thanks for your comment, the image has been modified as requested.

Fig 8 should not be qualitative. There is no way to tell when failure goes from Tensile to Shear. The axis ought to have a scale. Otherwise, Fig 8 is merely illustrative. There is no way to tell how much is high or slow.

Reply: Thanks for your comment. Figure 8 should be retained as it illustrates a trend that allows us to observe a link between abruptly changing loading rates and eventual failure modes, thus aiding our understanding of the manuscript.

I still think that a quantitative analysis of the stress field may shed a deeper insight on how the crack goes from Tensile to shear.

Suggested references (where the crack path has been adressed quatitatively):

Yates 2008. Crack paths under mixed mode loading. doi:10.1016/j.engfracmech.2007.05.014

Vormwald and Yang (2017)  Fatigue crack growth simulation under cyclic non-proportional mixed mode loading. DOI: 10.1016/j.ijfatigue.2017.04.014

Diaz and Freire (2021)  LEFM crack path models evaluation under proportional and non-proportional load in low carbon steels using digital image correlation data. DOI 10.1016/j.ijfatigue.2021.106687

Reply: Thanks for your comment. A quantitative analysis of the stress field is not included in our manuscript and will be our next work.
